# Modulation of Receptor Tyrosine Kinase Activity through Alternative Splicing of Ligands and Receptors in the VEGF-A/VEGFR Axis

**DOI:** 10.3390/cells8040288

**Published:** 2019-03-28

**Authors:** Megan Stevens, Sebastian Oltean

**Affiliations:** Institute of Biomedical & Clinical Sciences, Medical School, College of Medicine and Health, University of Exeter, Exeter EX1 2LU, UK

**Keywords:** VEGF, VEGFR, tyrosine kinase, alternative splicing

## Abstract

Vascular endothelial growth factor A (VEGF-A) signaling is essential for physiological and pathological angiogenesis. Alternative splicing of the VEGF-A pre-mRNA gives rise to a pro-angiogenic family of isoforms with a differing number of amino acids (VEGF-A_xxx_a), as well as a family of isoforms with anti-angiogenic properties (VEGF-A_xxx_b). The biological functions of VEGF-A proteins are mediated by a family of cognate protein tyrosine kinase receptors, known as the VEGF receptors (VEGFRs). VEGF-A binds to both VEGFR-1, largely suggested to function as a decoy receptor, and VEGFR-2, the predominant signaling receptor. Both VEGFR-1 and VEGFR-2 can also be alternatively spliced to generate soluble isoforms (sVEGFR-1/sVEGFR-2). The disruption of the splicing of just one of these genes can result in changes to the entire VEGF-A/VEGFR signaling axis, such as the increase in VEGF-A_165_a relative to VEGF-A_165_b resulting in increased VEGFR-2 signaling and aberrant angiogenesis in cancer. Research into this signaling axis has recently focused on manipulating the splicing of these genes as a potential therapeutic avenue in disease. Therefore, further research into understanding the mechanisms by which the splicing of VEGF-A/VEGFR-1/VEGFR-2 is regulated will help in the development of drugs aimed at manipulating splicing or inhibiting specific splice isoforms in a therapeutic manner.

## 1. Introduction

Angiogenesis comprises the formation and maintenance of blood vessels. A variety of signaling molecules are involved in the regulation of angiogenesis, including vascular endothelial growth factor (VEGF), which is essential both for physiological and pathological angiogenesis [1]. The biological functions of VEGF proteins are mediated by a family of cognate protein tyrosine kinase receptors, known as the VEGF receptors (VEGFRs) [2]. Activation of the VEGF pathway has been implicated in a large number of disease processes ranging from cancer to autoimmunity.

There are several VEGF proteins; VEGF-A binds to and signals through VEGFR-1 (Flt-1) and VEGFR-2 (KDR/Flk-1), VEGF-B signals solely through VEGFR-1, and VEGF-C and VEGF-D have a high affinity to VEGFR-3 (Flt-4) [1,2]. In addition, there are two neuropilin receptors, which are transmembrane glycoproteins, that function in the VEGF-VEGFR axis [2]; neuropilin-1 (NRP-1), a non-kinase co-receptor for VEGFR-2, functions to enhance the binding and signaling of certain isoforms of VEGF-A. NRP-2, on the other hand, is a non-kinase co-receptor for VEGFR-3. Since VEGFR-1 and VEGFR-2 are the receptor tyrosine kinases specific for VEGF-A, this review will focus on the splice variants of these two receptors only.

## 2. VEGFR Splice Variants and Functions

VEGF-A binds to two tyrosine kinase VEGFRs, VEGFR-1 and VEGFR-2. There are several isoforms of these VEGFRs that arise as a result of alternative splicing of the VEGFR pre-mRNA, which can alter the protein function, as detailed below (Figure 1). Both VEGFR-1 and VEGFR-2 have seven extracellular immunoglobulin (Ig)-like domains, which consist of a tetramer of two light chains and two heavy chains linked by disulphide bonds, a single transmembrane region, and an intracellular tyrosine kinase sequence interrupted by a kinase insert domain [3]. VEGF-A binds to the extracellular domain and the kinase insert domain acts as a binding site for intracellular proteins to carry out specific signaling cascades in response to ligand binding.

### 2.1. VEGFR-1 Signaling

VEGFR-1 was the first receptor tyrosine kinase for VEGF-A to be identified in COS cells [4] and has since been reported to be widely expressed on many cell types; however, it has very poor tyrosine kinase activity and is not required for endothelial cell function [5]. VEGFR-1 binds VEGF-A with high affinity but there is conflicting evidence for the role of VEGFR-1 as it appears to signal differently depending on the cell type and stage of development [5]. VEGFR-1 gene expression is regulated by hypoxia in human umblical endothelial cells; the VEGFR-1 promoter contains a binding site for hypoxia inducible factor (HIF)-1α [6]. Relatively little is known about the function of VEGFR-1. Constitutive knock-out (KO) of VEGFR-1 results in embryonic lethality between embryonic days 8.5 and 9 [7]. This was later found to be the result of increased endothelial cell outgrowth and angioblast commitment, which prevented proper organization of the vascular network [8]. Previous reports have labelled VEGFR-1 as a decoy receptor, decreasing the amount of VEGF-A readily available to bind to and phosphorylate VEGFR-2 [9]. Further evidence for this is that deletion of just the intracellular kinase domain for VEGFR-1 resulted in normal vascular development in mice [9]. Therefore, VEGFR-1 is hypothesized to sequester VEGF-A, preventing it from binding to its functional receptor, VEGFR-2.

### 2.2. Function of sVEGFR-1

The VEGFR-1 pre-mRNA can be alternatively spliced to produce the full-length membrane-spanning receptor described above, or the truncated soluble VEGFR-1 (sVEGFR-1), which includes the seven N-terminal immunoglobulin-like extracellular domains but not the transmembrane spanning or intracellular kinase domains, thus has a specific 31-amino acid c-terminus [1]. Full-length VEGFR-1 mRNA consists of 30 exons, whereas sVEGFR-1 only contains the first 13–14 exons due to intron retention and usage of an alternative polyadenylation signal and stop codon (isoforms detailed below). sVEGFR-1 is suggested to form non-signaling complexes with VEGFR-2, thus functioning as a modulator of VEGF-A signaling [10]. Like full length VEGFR-1, sVEGFR-1 has also been shown to act as a decoy receptor; VEGFR-1 KO mice die from vasuclar overgrowth due to increased signaling of VEGF-A through VEGFR-2; however, the administration of sVEGFR-1 to VEGFR-1 KO mice partially rescues this phenotype as it reduces the levels of VEGFR-2 phosphorylation [11].

There are currently five known VEGFR-1 protein coding isoforms (reviewed in [12]) (Figure 1A). Isoform 1 is denoted by the full-length VEGFR-1. Isoform 2 is termed sVEGFR-1, which comprises the 656 N-terminal residues followed by a specific 30-amino acid C-terminus and appears to have ubiquitous expression throughout most tissues [12]. Isoform 3 is a second soluble form generated by alternative splicing downstream of exon 14, termed sVEGFR-1_i14, which has been predominantly detected in the testes and brain [12]. Isoforms 4 and 5 result from the use of a new terminal exon, termed exon 15a and 15b, which is derived from an intronic sequence. These isoforms have been found to be highly expressed in the placenta [12]. Alternative splicing of VEGFR-1 involves *cis*-regulatory elements in the VEGFR-1 pre-mRNA within intron 13 [13]. Hypoxia is reported to increase the expression of transmembrane VEGFR-1 [6]; however, the effect of hypoxia on sVEGFR-1 expression is not so clear. In endothelial cells, hypoxia was shown to downregulate the expression of sVEGFR-1, which was not directly attributable to HIF-1α [14]. In contrast, exposure of macrophages/monocytes to granulocyte-macrophage colony-stimulating factor (GM-CSF) under hypoxic conditions results in HIF-2α-dependent changes in sVEGFR-1 expression [15]. In cytotrophoblasts, where the sVEGFR-1_i14 isoform is most commonly expressed, hypoxia increases both sVEGFR-1_i14 and sVEGFR-1 mRNA, which is proposed to be through HIF-1α [16]. Furthermore, sVEGFR-1_i14 secretion was shown to increase under hypoxic conditions through activation of the growth arrest and DNA damage-inducible 45a (Gadd45a) factor and p38 phosphorylation [17]. Several drugs and protein factors have been shown to modulate sVEGFR-1 expression, including Jumonji domain-containing protein 6, which interacts with the splice factor U2AF65 resulting in augmented levels of sVEGFR-1 in hypoxic conditions [18]. In addition, hnRNP D and arginine methylation have also been reported to play important roles in the regulation of sVEGFR-1 mRNA alternative polyadenylation [19]. Interestingly, VEGF-A can increase the expression of sVEGFR-1 through VEGFR-2-dependent activation of protein kinase C [20].

### 2.3. VEGFR-2 Signaling

VEGFR-2 is the main signaling receptor for VEGF-A. It is primarily located on endothelial cells and is essential for endothelial cell biology both during development and during physiological and pathological processes in adults. Like VEGFR-1, all VEGF-A isoforms contain residues that enable them to bind to VEGFR-2 and all bind with the same affinity. However, the affinity of VEGF-A for VEGFR-2 is 10-fold lower than that for VEGFR-1 [21,22]. A constitutive KO of VEGFR-2 results in embryonic lethality on day 8.5–9.5; mice lack mature endothelial and hematopoietic cells [23]. This is similar to the phenotype observed in VEGF-A KO mice [24]. Therefore, unlike VEGFR-1, VEGFR-2 signaling is crucial for vascular development.

Proteolytic hydrolysis of membrane-bound VEGFR-2 results in the generation of soluble VEGFR-2 (sVEGFR-2) [12]. sVEGFR-2 is proposed to function as an inhibitor of angiogenesis by binding to and sequestering VEGF-A, blocking canonical VEGF-A-VEGFR-2 signaling [25,26]. A further sVEGFR-2 isoform generated by intron 13 retention has been described; as with VEGFR-1, retention of intron 13 yields a truncated transcript whose protein variant lacks the transmembrane and intracellular kinase domain of full length VEGFR-2 [27] (Figure 1B). This splice variant is reported to play a role in lymphangiogenesis by blocking VEGF-C [27]. Little is known regarding the mechanisms controlling this alternative splicing event.

## 3. VEGF-A Splice Variants

The human VEGF-A pre-mRNA consists of eight exons and seven introns. Alternative splicing of the VEGF-A pre-mRNA gives rise to a family of isoforms with a differing number of amino acids due to the exclusion/inclusion of various exons (e.g., VEGF-A_121_, VEGF-A_165_, VEGF-A_189_, and VEGF-A_206_, collectively known as VEGF-A_xxx_a where xxx denotes the number of amino acids) (Figure 2). Such isoforms are widely known to be pro-angiogenic, pro-permeability factors. In addition, the selection of an alternative 3′ splice site, known as the distal splice site, in exon 8 of the VEGF-A pre-mRNA results in a new family of VEGF-A isoforms, termed VEGF-A_xxx_b [28]. The resulting VEGF-A_xxx_b proteins differ in the C-terminal sequence by only six amino acids, resulting in radically different functional properties (Figure 2). In comparison to VEGF-A_xxx_, VEGF-A_xxx_b isoforms are collectively anti-angiogenic and reduce vessel permeability (anti-permeability). Sixteen isoforms of VEGF-A have been identified, including an additional isoform, VEGF-Ax, which arises from translational read-through of the VEGF-A transcript beyond the canonical stop codon (programmed translational read-through) [29].

VEGF-A splicing is predominantly regulated by a group of RNA binding proteins known as serine/arginine (SR) proteins. SRSF1, SRSF2, SRSF5, and SRSF6 have all been reported to play a role in VEGF-A alternative splicing [30]. Upon phosphorylation of multiple serine/arginine and proline/serine repeats, SR proteins are translocated from the cytoplasm to the nucleus where they bind to exonic sequence enhancers within the VEGF-A pre-mRNA, resulting in the splicing out of an exon [31]. The inclusion/exclusion of certain exons result in the different isoform properties of each VEGF-A protein. Exons 1–5 are constitutive exons; they encode a single sequence (exons 1/2), a glycosylation site (Asp74), a potential plasmin cleavage site (Arg110 and Ala111), as well as VEGFR binding residues [32,33]. Whereas exons 1–5 are present in all isoforms of VEGF-A, exons 6 and 7 are alternatively spliced. Heparin sulfate (HS) glycoproteins are present in the extracellular matrix (ECM) and can interact with both VEGF-A and VEGFRs, thus they are suggested to regulate the bioavailability of VEGF-A. Residues in exon 6a and 7 of VEGF-A are responsible for the interaction with HS [34]. VEGF-A_145_, VEGF-A_189_, and VEGF-A_206_ all contain exon 6a and 7 resulting in a high affinity for HS; this results in these longer isoforms being tethered to the ECM. On the other hand, VEGF-A_111_ and VEGF-A_121_ lack exon 6 and 7, so they are unable to bind HS, making them freely diffusible in the ECM and more bioavailable [35]. The most dominant isoform is VEGF-A_165_, which contains exon 7 but not 6. Therefore, VEGF-A_165_ has an intermediate bioavailability as approximately 50% remains cell- or ECM-bound [36].

Regarding exon 8 of the VEGF-A gene, selection of either the proximal or distal splice site has been reported to be dependent on the type of external stimulus; proximal splice site selection is promoted by insulin like growth factor (IGF1) and tumor necrosis factor alpha (TNFα), whereas distal splice site selection is promoted by tumor growth factor beta 1 (TGF-B1) [37]. A widely reported example of exon 8 splicing regulation involves serine/threonine-protien kinase 1 (SRPK1) and CDC-like kinase 1 (Clk-1). SRPK1 activation has been shown to phosphorylate SRSF1, resulting in proximal splice site selection and the translation of VEGF-A_xxx_a proteins [38]. On the other hand, Clk-1 signaling results in the phosphorylation of SRSF6, with the distal splice site being subsequently selected and VEGF-A_xxx_b proteins translated [37]. Other reported regulators of VEGF-A exon 8 splicing are E2F1 and SRSF2, which were both shown to increase the VEGF-A_xxx_b/VEGF-A_xxx_a ratio [39].

## 4. VEGFR Signaling

### 4.1. Role of VEGFR-1 Signlaing and sVEGFR-1 Isoforms

As mentioned previously, the role of VEGFR-1 in vasculogenesis and angiogenesis has been ascribed to VEGF-A binding, thus regulating the amount of VEGF-A available for vascular development. VEGFR-1 is widely expressed but has poor kinase activity and is not required for endothelial cell function. Further evidence for this hypothesis arose from mice with a homozygous deletion of the VEGFR-1 tyrosine kinase domain developing healthy vasculature [9]. Therefore, the primary role of VEGFR-1 in embryonic angiogenesis is restricted to its extracellular region and is independent of its tyrosine kinase activity. As sVEGFR-1 contains the extracellular domain, it also acts as a decoy receptor [40]. sVEGFR-1 is also proposed to form non-signaling complexes with VEGFR-2 [10].

A study using VEGFR-1 KO embryonic stem cells showed that sVEGFR-1 is important for the modulation of endothelial cell migration and vascular sprouting during development [41]. During vessel morphogenesis, endothelial cells are suggested to form a VEGF-A gradient via the interaction of VEGF-A with sVEGFR-1, resulting in sequestration of VEGF-A and local inactivation of VEGFR-2 signaling [42]. Therefore, sVEGFR-1 is proposed to act as a guidance molecule during vessel sprouting, i.e., inactivating VEGF-A either side of the sprout to provide a VEGF-A-rich corridor for the emerging vessel [43]. sVEGFR-1 present in the ECM is also reported to play a role in α5β1 integrin signaling regarding the cell adhesion pathway [44]; however, these signaling pathways are not related to VEGF-A and are beyond the scope of this review.

Recent studies have highlighted that VEGF-B and PIGF are able to signal through VEGFR-1, eliciting a pro-angiogenic effect independent of VEGF-A [45,46]. In addition, increased levels of sVEGFR-1 have been observed in vascular pathologies [45], indicating that VEGFR-1 may act as more than a decoy receptor/VEGFR-2 inhibitor.

The role of sVEGFR-1 in tumor development and progression has been widely reported. The expression of sVEGFR-1 has been found to be increased in many types of cancer, including glioblastoma, melanoma, breast, hepatocellular, lung, leukemia, colorectal, renal, and head and neck [47,48,49,50,51,52,53,54,55]. Increased circulating sVEGFR-1 is often correlated with poor prognosis; however, the balance between VEGF-A and sVEGFR-1 may be more important when considering the clinical outcome. For example, increased sVEGFR-1 and VEGF-A are correlated with poor prognosis in lung cancer patients [51]. On the other hand, increased VEGF-A combined with low levels of sVEGFR-1 are associated with a poor prognosis in breast cancer [56]. In addition to being a marker for tumor progression, sVEGFR-1 has also been shown to serve as a biomarker for tumor response to therapy. Using the example of bevacizumab, increased plasma levels of sVEGFR1 were reported to be inversely correlated with treatment response in breast cancer [57]. However, this appears to be dependent on the type of cancer as the sVEGFR-1 expression level was found to be decreased upon treatment of metastatic colorectal cancer [58].

Excess circulating soluble isoforms of VEGFR-1 have been shown to contribute to the pathogenesis of pre-eclampsia in pregnant women [59,60]. The sVEGFR-1_i14 isoform is presumed to be a major contributor to this condition because it is selectively expressed by placental cytotrophoblasts; the increased sequestration of platelet-derived growth factor (PIGF) and VEGF-A by excess sVEGFR-1_i14 results in endothelial dysfunction and altered neutrophil activation and migration, ultimately causing hypertension, proteinuria, and glomerular endotheliosis in patients [60,61]. Indeed, increased levels of circulating sVEGFR-1_i14 is used as a biomarker for the development of pre-eclampsia [62].

As described above in pregnant women with pre-eclampsia, increased circulating levels of sVEGFR-1 is linked to endothelial dysfunction in the glomeruli of the kidney. VEGF-A is secreted by the glomerular epithelial cells (podocytes) to signal to VEGFR-2 on the glomerular endothelial cells, a process that is tightly regulated to maintain proper functioning of the glomerular filtration barrier. Plasma levels of sVEGFR-1 are higher in patients with chronic kidney disease (CKD), which are correlated with cardiovascular disease [63,64]. On the other hand, inducible over-expression of podocyte sVEGFR-1 has been shown to be therapeutic in a model of diabetic nephropathy where excess VEGF-A expression is observed [65]. In addition, sVEGFR-1 has been reported to bind to lipid microdomains in podocytes, which can alter cell morphology and the function of the glomerular filtration barrier [66].

sVEGFR-1 has also been shown to play a role in ocular pathologies through the inhibition of VEGF-A, including the preservation of cornea avascularity [67]. In addition, reduced levels of sVEGFR-1 where observed in patients with age-related macular degeneration [68]. Regarding inflammation, increased levels of sVEGFR-1 in the blood is indicated to act as a potential new biomarker of sepsis [69], and a predictor of endothelial dysfunction/activation of coagulation in acute pancreatitis [70].

On the other hand, in mouse xenograft models of melanoma, lung cancer, fibrosarcoma, and glioblastoma, exogenous administration of sVEGFR-1 (either transfection, recombinant protein, or adenovirus infection) inhibited tumor growth and neoangiogenesis, increasing the survival rate [71,72,73,74].

### 4.2. VEGF-A_xxx_b Activation of VEGFR-1

Information on VEGFR-1 activation and signaling is sparse; however, a recent study has shown that VEGF-A_165_b inhibits VEGFR-1 signaling in ischemic muscle in mice, and that VEGF-A_165_b inhibition induces activation of VEGFR-1 [75]. Furthermore, in vitro studies showed that VEGF-A_165_b failed to induce the activation of VEGFR-1-Y1333, reducing VEGFR-1-STAT3 signaling [75].

### 4.3. Mehcanisms of VEGFR-2 Signaling

As mentioned above, all VEGF-A isoforms can bind to VEGFR-2 with similar affinity; however, different isoforms result in different activation and signaling outcomes [32] (Figure 3). Upon binding of VEGF-A to its orthosteric ligand binding site, VEGFR-2 undergoes dimerization and a conformational twist in the extracellular region results in the rotation of transmembrane helices [76,77]. Both VEGF-A_165_ and VEGF-A_165_b have been shown to result in VEGFR-2 dimerization [77]. Conformational changes in the intracellular domain of VEGFR-2 follows; ATP binds to the flexible N-lobe cleft facilitating the intrinsic kinase activity of the receptor and phosphorylation of the tyrosine residues in the C-lobe [78]. Upon phosphorylation of these tyrosine residues, certain cytoplasmic proteins bind and distinct signaling pathways are initiated, included those involved in cell survival, migration, proliferation, vasodilatation, and permeability (reviewed in [79]). The tyrosine residues include Y1054 and Y1059 in the activation loop, which are required for maximal kinase activity of VEGFR-2 [80]; Y951 in the kinase insert domain, which serves as a binding site for T cell-specific adapter molecule (TSAd) [81], and is vital for HUVEC migration in response to VEGF-A [82]; and Y1175 and Y1214 in the COOH-terminal tail. Y1175 phosphorylation mediates cell proliferation through binding of phospholipase C (PLC)-γ [83]. VEGFR-2 is dephosphorylated by protein phosphatase 1b (PTP1b) in the endoplasmic reticulum, which highlights the importance of spatiotemporal trafficking on the activation of VEGFR-2 [84,85].

### 4.4. VEGFR-2 Signaling in Angiogenesis

During sprouting angiogenesis, endothelial cells within existing vessels form an angiogenic sprout towards a chemotactic stimulus, such as VEGF-A. The angiogenic sprout is orientated with a leading tip cell and trailing stalk cells. The extent of sprouting in neighboring endothelial cells is regulated by delta-like ligand 4 and Notch via lateral inhibition [86]. Lumen formation occurs once two sprouts anastomose, and the new vessel is stabilized by smooth muscle cell and basement membrane deposition [87].

Cell proliferation is required for angiogenesis. VEGF-A activates VEGFR-2 and stimulates proliferation through the activation of RAS, which then activates RAF kinase to phosphorylate mitogen-activated protein kinases (MAPK/ERK) [88]. VEGFR-2 stimulates ERK activation via Y1175-dependent phosphorylation of PLC-γ, resulting in the subsequent activation of protein kinase C (PKC) [82]. Mutation of Y1175 or administration of and antibody specific to Y1175 decreased VEGF-A-dependent cell proliferation in vitro [89]. Furthermore, mutation of Y1175 in mice results in embryonic lethality on day 5–9 due to a lack of blood vessel formation [90].

Endothelial cell migration is also essential for angiogenesis. One VEGFR-2 signaling pathway that has been implicated in endothelial cell migration is initiated via the phosphorylation of Y951, which allows for the binding of T cell specific adapter protein (TSAd) [81]. Both mutation of Y951 and knock-down of TSAd are reported to inhibit VEGF-A-mediated actin reorganization, and thus migration in cultured endothelial cells; however, proliferation remained unaffected [81]. Another example of a VEGFR-2 signaling pathway involves phosphorylation of Y1175 to induce focal adhesion kinase (FAK)-mediated endothelial cell migration [91].

### 4.5. VEGFR-2 Signaling in Cell Survival

VEGF-A activation of VEGFR-2 is associated with increased endothelial cell survival. VEGFR-2 activates phosphoinositide 3-kinase (PI3K), which enables membrane recruitment and phosphorylation of protein kinase B (PKB/AKT) [92]. Activation of the cell survival factor AKT results in the phosphorylation of Bcl-2 associated death promoter (BAD), inhibiting the activity of pro-apoptotic factors such as Bcl-2 and caspase 9 [93].

### 4.6. VEGFR-2 Signaling in Permeability

VEGF-A activation of VEGFR-2 induces extravasation of proteins and leukocytes in vivo [94]. This is suggested to occur through two mechanisms: the formation of transcellular endothelial pores and the transient opening of paracellular junctions [95]. However, the exact signaling mechanisms regulating these events are not yet clear. One suggested mechanism involves VEGF-A-dependent endothelial nitric oxide synthase (eNOS) activation through PLC-γ and AKT, resulting in the activation of the pro-permeability factor nitric oxide (NO) [96,97].

### 4.7. Role of sVEGFR-2

The alternatively spliced sVEGFR-2 isoform has been reported to act as an endogenous VEGF-C antagonist, preventing it from binding to VEGFR-3 and consequently inhibiting lymphatic endothelial cell proliferation [27]. In addition, like sVEGFR-1, sVEGFR-2 is a natural circulating decoy receptor for VEGF, thus acting as a ligand trap [98].

### 4.8. VEGF-A Isoform Specific Activation of VEGFR-2

The canonical VEGF-A_xxx_a isoforms are widely described as pro-angiogenic, pro-permeability factors as they activate the aforementioned signaling pathways via VEGFR-2 binding and dimerization. On the other hand, VEGF-A_xxx_b isoforms are anti-angiogenic and anti-permeability, which is due to their effect on VEGFR-2 activation. Like VEGF-A_xxx_a, VEGF-A_xxx_b is still able to bind and dimerize VEGFR-2, but whether they result in phosphorylation of the tyrosine residues in the intracellular domain is not clear. The six-amino acid frame shift that occurs when the distal splice site is selected in the VEGF-A pre-mRNA results in the replacement of a positively charged arginine residue with neutral aspartic acid and lysine, which are predicted to decrease VEGFR-2 activation [99]. In pulmonary arterial endothelial (PAE) cells, VEGF-A_165_b was shown to induce VEGFR-2 activation (Y1052, Y1057) compared to untreated controls, but not to the same extent as that induced by VEGF-A_165_ [99]. Another report suggested that recombinant VEGF-A_165_b can induce Y1175 activation to almost the same extent as VEGF-A_165_ in HEK293-VR2 cells [100]. In addition, VEGF-A_165_b can induce VEGFR-2 Y1175 phosphorylation to the same extent as VEGF-A_165_ in endothelial cells [75]. However, anti-VEGF-A_165_b treatment of HUVECs and cultured visceral adipose tissue resulted in increased Y951 phosphorylation [101,102], indicating that VEGF-A_165_b antagonized Y951 phosphorylation. Furthermore, treatment of glomerular endothelial cells with VEGF-A_165_b did not result in any increases in the overall phosphorylated state of VEGFR-2 (immunoprecipitation of VEGFR-2 followed by immunoblotting with a phospho-tyrosine antibody) [103]. Taken together, these findings indicate that VEGF-A_165_b acts as a VEGFR-2 partial agonist/antagonist via the differential modulation of site-specific phosphorylation on VEGFR-2.

In some pathologies, VEGF-A_165_b expression has been shown to be down-regulated relative to VEGF-A_165_a. For example, in the late stages of human diabetic nephropathy when the kidney is not filtering properly, kidney VEGF-A_165_b levels are down-regulated relative to VEGF-A_165_a; however, during the early stages of diabetic nephropathy when the kidney is functioning well, the VEGF-A_165_b isoform is increased [104]. Therefore, VEGF-A_165_b may play a protective role in early nephropathy but when the expression is decreased, increased angiogenesis and permeability occur, resulting in a worse phenotype. Indeed, several studies in mouse models have shown the VEGF-A_165_b isoform to have reno-protective effects regarding glomerular permeability [103,104,105,106]. These protective effects are indicated to be due to VEGF-A_165_b decreasing the phosphorylation of VEGFR-2, which has been shown in glomerular endothelial cells [103]. Decreased levels of VEGF-A_165_b have also been observed in certain cancers, including colon cancer and renal cell carcinoma [28,107]. This reduction in VEGF-A_165_b is often accompanied by an increase in the pro-angiogenic VEGF-A_165_a, which contributes to angiogenesis within the tumor. Administration of VEGF-A_165_b, or manipulation of VEGF-A splicing to promote VEGF-A_165_b expression (such as with SRPK1 inhibitors), has been shown to be therapeutic in many tumor models through inhibition of VEGF-A_xxx_a mediated angiogenesis [108,109]. On the other hand, VEGF-A_165_b has also been shown to promote lung tumor progression and specific knock-down of just the VEGF-A_165_b isoform reduced tumor growth in lung cancer cells [110]. Thus, the role of VEGF-A_165_b signaling may depend on the tissue it is expressed in.

VEGF-A_121_a is a shorter freely diffusible VEGF-A isoform. In contrast to VEGF-A_165_a, VEGF-A_121_a has been shown to exhibit both partial and full agonist effects. On one hand, VEGF-A_121_a acts as a partial agonist of VEGFR-2 in both in vivo and in vitro measurements of angiogenesis and signaling, respectively [5,99], as well as slowing HUVEC proliferation and reducing sprouting in comparison to VEGF-A_165_a [111,112]. In contrast, VEGF-A_121_a-induced angiogenic sprouting ex vivo has been reported to be both comparable [33] and reduced [113] in comparison to VEGF-A_165_a. Similar trends are seen regarding vascular permeability [114,115,116].

VEGF-A_145_a and VEGF-A_189_a are ECM-bound isoforms that also show reduced agonistic effects on VEGFR-2 signaling in comparison to VEGF-A_165_a. In HUVECs, VEGF-A_145_a had a reduced effect on proliferation and permeability relative to VEGF-A_165_a, but comparable effects on migration [114]. This was indicated to be due to reduced phosphorylation of VEGFR-2 in addition to reduced activation of AKT and ERK [114]. Similarly, VEGF-A_189_a resulted in decreased cell survival and proliferation in BAECs, but comparable effects to VEGF-A_165_a on migration [117,118].

## 5. VEGFR Signaling Complexes

### 5.1. VEGFR Heterodimerization

Computational modeling has predicted VEGFR-1/2 heterodimers to comprise 10–50% of signaling VEGFR complexes, which are favored over VEGFR-1 homodimers when the VEGFR-2 abundance is higher [119]. There is evidence that suggests that VEGF-A stimulation of VEGFR-2 homodimers, VEGFR-1 homodimers, and VEGFR-1/2 heterodimers results in different efficacies of signal transduction; the pattern of Ca2+ flux was found to be unique for each type of receptor dimer in porcine aortic endothelial cells [120]. VEGF-A, VEGF-C, and VEGF-D have also been shown to induce the heterodimerization of VEGFR-2/3, which is required for certain ligand-dependent cellular responses mediated by VEGF-C and VEGF-D [121].

### 5.2. Roles of Neuropilins NRP1 and NRP2

Neuropilins can function as coreceptors with VEGFR-1 and VEGFR-2. There are two homologs of NRP, NRP1 and NRP2, which consist of a single transmembrane spanning domain with a small cytoplasmic domain lacking intrinsic catalytic function [122]. NRP1 was firstly suggested to bind in exon 7 of VEGF-A, which is present in isoforms such as VEGF-A_165_, forming a ternary complex with VEGFR-2 [112], thus primarily acting as a co-receptor for VEGFR-2. More recent studies have implicated the exon 8a-encoded arginine residue in the binding of VEGF-A to the b1 domain of NRP1 [123]. Binding of VEGF-A to NRP1 enhances VEGF-A signaling in endothelial cells with respect to migration and survival [124,125,126]. Furthermore, NRP1 is reported to be essential for VEGF-A-induced vessel sprouting and branching in angiogenesis [127]. NRP1 has also been shown to be associated with the adapter Synectin (GIPC), which is associated with the intracellular trafficking of VEGFR-2 [125]. In contrast, NRP2 acts as a co-receptor for VEGFR-3 and is therefore not involved with VEGF-A signal transduction [128]. In mice, both overexpression and disruption of NRP1 results in embryonic lethality on E12.5-13.5 due to vascular abnormalities [129]. Furthermore, siRNA [113] or antibody [112] blocking of NRP1 led to a decrease in VEGF-A_165_a-induced phosphorylation of VEGFR-2 in vitro.

In contrast to VEGF-A_165_, VEGF-A_189_, and VEGF-A_145_, fluorescent real-time ligand binding assays revealed that VEGF-A_165_b and VEGF-Ax are unable to bind to NRP1 as they lack the exon 7-8a-encded residues [130]. This provides further evidence for the lack of VEGFR-2 singling induced by the weak agonist VEGF-A_xxx_b isoforms. There is conflicting data regarding the binding of VEGF-A_121_a to NRP1 as it lacks exon 7, with most studies suggesting that although VEGF-A_121_a can bind NRP1, albeit at a lower affinity, it is unable to bridge the NRP1/VEGFR-2 complex (reviewed in [131]).

### 5.3. NRP1 and NRP2 Splice Variants

NRP1 exists as a full-length membrane-bound form in addition four soluble isoforms. Full-length NRP1 is comprised of 17 exons. On the other hand, two soluble splice variants, s_12_NRP1 and s_11_NRP1, are generated during pre-mRNA processing via intron read-through in the NRP1 gene, resulting in proteins that lack transmembrane and cytoplasmic domains of full-length NRP1 [132,133]. Functionally, these soluble isoforms of NRP1 were reported to bind VEGF-A_165_, although not VEGF-A_121_, thus inhibiting VEGF-A_165_-induced phosphorylation of VEGFR-2 in endothelial cells resulting in reduced tumor growth (anti-tumor properties) [133]. Therefore, s_12_NRP1 and s_11_NRP1 appear to act as VEGF-A_165_ antagonists. Two further soluble isoforms of NRP1 have also been described, s_III_NRP and s_IV_NRP, which are proposed to have similar biological and biomechanical properties as s_12_NRP1 and s_11_NRP1 [134]. The s_III_NRP1 isoform results from the deletion of exons 10 and 11, while exon 12 is still present, followed by retention of the beginning of intron 12 (28 bp). The s_IV_NRP1 isoform is missing exon 11, also resulting in intron 12 retention [134]. Both s_III_NRP and s_IV_NRP have been shown to be expressed in normal and cancerous tissues and are capable of binding VEGF-A_165_, indicating that these two isoforms are antagonists for NRP1-mediated cellular activities [134]. The final isoform of NRP1 is NRP∆E16, which results from the skipping of exon 16 and replacement with an “AAG” Arg triple; however, this isoform does not have a functional difference to full length NRP1 [135].

NRP2 can also exist as a membrane bound or soluble form. The membrane bound form of NRP2 has two splice variants, NRP2a and NRP2b, which differ in the last 100 amino acids of the c-terminus. Therefore, these two splice variants are proposed to bind different proteins and govern different molecular pathways [132]. NRP2b has been reported to have a prometastatic role in non-small cell lung cancer, whereas NRP2a in promoting metastasis and therapy resistance [136]. However, further studies are needed to clarify the roles of each of these splice variants with respect to VEGF-A binding and signaling.

## 6. Regulation of Splicing as a Therapeutic Intervention

Research into the VEGF-A-VEGFR signaling axis in disease has recently taken a new direction focused on manipulating the splicing of these genes as a potential therapeutic avenue. One example of this is the regulation of the VEGF-A_xxx_a/VEGF-A_xxx_b ratio. Small molecule inhibitors of SRPK1, known as SRPIN340 and SPHINX31, have been shown to upregulate the VEGF-A_xxx_b isoforms relative to VEGF-A_xxx_a, which had a therapeutic effect in animal models of retinopathy [137,138]. Furthermore, a natural blueberry extract as also been shown to increase VEGF-A_165_b/VEGF-A_164_a in the kidneys of diabetic mice, exerting a therapeutic effect through a decrease in kidney fibrosis and permeability [139]. Regarding the VEGFRs, exogenous administration of sVEGFR-1 (either transfection, recombinant protein, or adenovirus infection) was reported to inhibit tumor growth and neoangiogenesis, increasing the survival rate in mouse xenograft models of melanoma, lung cancer, fibrosarcoma, and glioblastoma [71,72,73,74]. Therefore, further research into the regulation of VEGFR splicing is warranted to explore the potential therapeutic benefits of switching VEGFR splicing.

## 7. Conclusions

The VEGF-A-VEGFR axis is critical in both physiological and pathological angiogenesis and vessel permeability. The disruption of the splicing of just one of the genes involved in the VEGF-A-VEGFR axis (VEGF-A, VEGFR-1, VEGFR-2) can result in changes to the entire signaling axis, such as the increase in VEGF-A_165_a relative to VEGF-A_165_b resulting in increased VEGFR-2 signaling and aberrant angiogenesis in cancer. Further research into understanding the mechanisms by which the splicing of VEGF-A/VEGFR-1/VEGFR-2 is regulated will help in the development of drugs aimed at manipulating splicing or inhibiting specific splice isoforms in a therapeutic manner.

## Figures and Tables

**Figure 1 cells-08-00288-f001:**
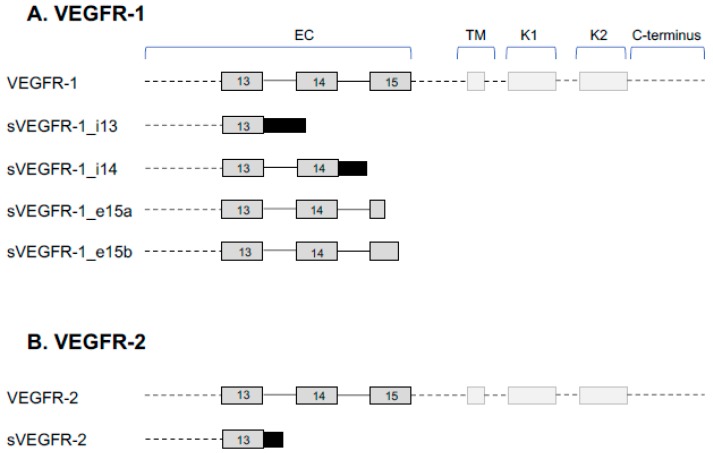
Alternative splice variants of VEGFR-1 and VEGFR-2. (**A**) Alternative splicing gives rise to five known splice variants of VEGFR-1: full length VEGFR-1, intron 13 retention (sVEGFR-1_i13), intron 14 retention (sVEGFR-1_i14), terminal exon 15a (sVEGFR-1_e15a), and terminal exon 15b (sVEGFR-1_e15b). The soluble isoforms only contain the extracellular (EC) domain and are missing the transmembrane (TM) and kinase (K1 and K2) domains. (**B**) Alternative splicing gives rise to two known splice variants of VEGFR-2: full length VEGFR-2 and sVEGFR-2, which result from intron 13 retention. The sVEGFR-2 only contains the EC domain.

**Figure 2 cells-08-00288-f002:**
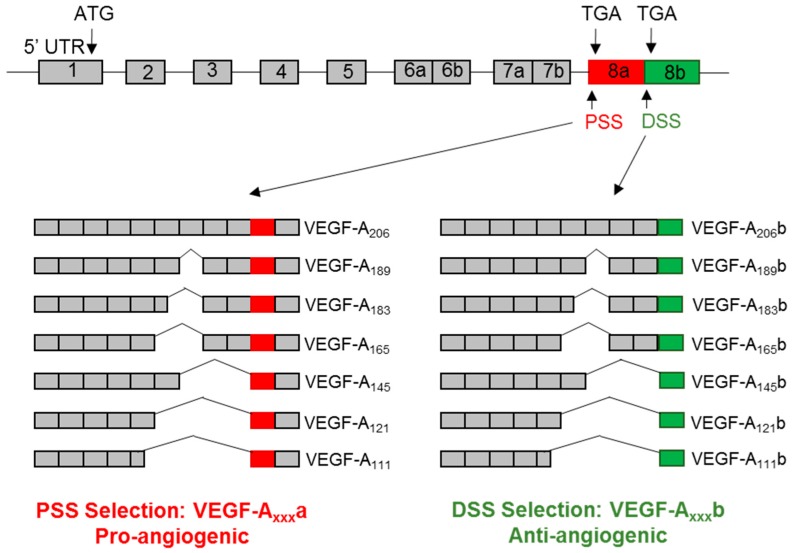
Alternative splicing of VEGF-A. The VEGF-A pre-mRNA is comprised of 8 exons. Inclusion/exclusion of exons 6a,b and 7a,b gives rise to VEGF-A isoforms with differing numbers of amino acids. The use of an alternative 3′ splice site in exon 8 results in a differing c-terminal sequence of amino acids (VEGF-A_xxx_b isoforms). The VEGF-A_xxx_a family of isoforms have pro-angiogenic, pro-permeability properties whereas the VEGF-A_xxx_b isoforms are anti-angiogenic and anti-permeability. Figure adapted from Stevens et al. 2018.

**Figure 3 cells-08-00288-f003:**
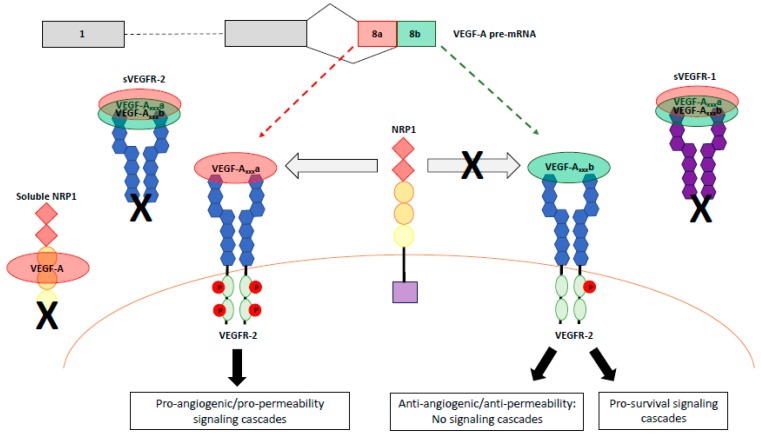
VEGF-A_xxx_a and VEGF-A_xxx_b signaling through VEGFR splice variants and NRP1. Both VEGF-A_xxx_a and VEGF-A_xxx_b can bind and dimerize VEGFR-2. VEGF-A_xxx_a recruits NRP1, a co-receptor for VEGFR-2, which results in phosphorylation of the tyrosine kinase domains of VEGFR-2, producing pro-angiogenic and pro-permeability intracellular signaling cascades. In contrast, VEGF-A_xxx_b is unable to recruit NRP1, resulting in weak, transient phosphorylation of VEGFR-2 and some pro-survival signaling cascades. Soluble isoforms of NRP1, as well as sVEGFR-2 and sVEGFR-1 lack transmembrane domains and act as decoy receptors, sequestering VEGF-A.

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
