# Peer review of "Modulation of Receptor Tyrosine Kinase Activity through Alternative Splicing of Ligands and Receptors in the VEGF-A/VEGFR Axis"

_cells, 2019, doi:10.3390/cells8040288_

Round 1

Reviewer 1 Report

Summary / significance:

The role of VEGF-A and its interaction with according receptors (VEGFR-1 and VEGFR-2) is matter of extensive research in a wide range of diseases that do not necessarily involve vessel growth. This review by Stevens & Oltean focuses on current articles highlighting recent findings contributing to the understanding of VEGF signaling modulation through splice variant formation of VEGF-A and its receptors VEGFR-1 and VEGFR-2. The authors give an introductory overview of the VEGF-A growth factor and according VEGF receptor subtypes. They particularly point out the abundance of VEGF / VEGFR isoforms in diverse settings and conclude that these subtle differences are essential understand its mode of action.

Level of interest/merit:

Understanding of VEGF-A and VEGFR isoform abundance and function is of high interest to the scientific readership to develop novel powerful tools for use in anti- and/or pro-angiogenic therapy. This is an interesting review focussing on studies that investigated VEGF signaling regulation in various contexts.

There are some comments:

This article is written clearly, the contents are accurately and coherent, and the English level is very good. Several points could be addressed to make the text much more exciting and fluent to read:

The introductory part could be improved by giving more general and comprehensive information of VEGF-A, VEGFRs, neuropilins and the induced signaling path in the context of diverse tissue and organ settings. The first paragraph immediately goes too much into detail of listing signalling members, instead of attracting attention to the importance of this growth factor pathway. Evidence should be underscored by respective citations of additional key research papers / reviews. Maybe use text blocks included in subsequent sections of the manuscript.

The second paragraph leads over to the splice variants of VEGFR-1 and VEGFR-2. The beginning of this section could be kept more introductory and general, maybe also by improving Figure 1

(e.g. include information about Ig-like domains).

Overall, the sub-headings and flow of the subsections could be improved by including more descriptive text in the headlines.

The very extensive “VEGR-1” section (line 160f) could be split into two parts “VEGFR-1 signaling” and “function of soluble sVEGR-1”.

An overall summary and/or hints on the implications of the research for clinical application is not provided. Examples for therapeutic intervention with VEGF-A and VEGFR-splicing are missing, maybe drawing the attention to recent innovations, underlined by reviews.

Detailed comments:

Line 26f: please include respective citations throughout the whole first paragraph.

Line 29: be more explicative about the neuropilins “there are co-receptors in this axis…”

Line 35: maybe more extensive title such as “ VEGFR splice variants and functions”

Line 36f: refer to the protein structure shown in Figure 1.

Lines 41 and 92: maybe more extensive heading than just “VEGFR-1” and “VEGFR-2”?

Line 42: incomplete introductory sentence, which cell types / tissues? add according references.

Line 46: is the regulation by hypoxia ubiquitous in all tissues?

Line 51: unclear explanation, what is VEGFR-2 function? What does that mean for VEGFR-1 function?

Line 53: please give a brief summary of what is the hypothesized overall role of VEGR-1.

Line 55: conflict to line 36, where the authors mention seven Ig-like domains.

Line 58: unclear description of splicing process. What does mean “following intron 13 retention”?

Do you mean an exon 13+14 fusion?

Line 62: functioning of VEGFR-1 plus VEGFR-2 needs more explanation to understand effect on mouse rescue phenotype / reduction of VEGFR-2 phosphorylation.

Line 64: “there are currently five isoforms known

Line 69: maybe a brief hint, where these isoforms have been detected?

Line 71: “fully expressed transmembrane VEGFR-1”

Line 85f: legend to Figure 1: maybe better “alternative splice variants of VEGFR-1 and VEGFR-2”.

Line 89: B) instead of 2)

Line 102: include according reference, which proteolytic cleavage?

Line 120: what means “anti-permeability”? please use better description.

Line 122: please briefly explain what is meant with “translational read-through”.

Line 132: “upon phosphorylation of multiple se/arg repeats…”

Line 140: “…, thus they are suggested to regulate….”

Line 143: “…results in the longer isoforms…”

Line 144: “…lack exon 6 and 7, so they are unable to bind HS,…”

Line 148: “…regarding exon 8 of the VEGF-A gene, selection of either the proximal or the distal splice site has been reported to be dependent on the type of external stimulus.”

Line 152: please explain the full names of SRPK1 and Clk-1.

Lines 160: longer subtitle? Maybe “role of VEGFR-1 signaling and sVEGFR-1 isoforms”?

Line 161: “as mentioned above,…”

Line 228: longer subtitle? Maybe “mechanism of VEGFR-2 signaling”?

Line 291: longer subtitle? Maybe “role of sVEGFR-2”?

Line 300: “…isoforms act anti-angiogenic and anti-permeable…”

Line 354: “…when the VEGFR-2 abundance is higher”

Line 355: “there is evidence that suggests that..”

Line 362: better subtitle maybe “roles of neuropilins NRP1 and NRP2”

Line 363f: maybe better introductory sentence: “neuropilins can function as co-receptors of VEGFA, VEGFR-1 and VEGFR-2.”

Line 385: better “NRP1 and NRP2 splice variants”?

Line 388: “…in the NRP1 gene, resulting in proteins that….”

Line 392: what do you mean with “anti-tumor properties”?

Line 404: “…and govern different molecular pathways as a result.”

Line 407: “…with respect to VEGF-A binding and signaling”

Line 415f: this part of the conclusion is just repetition of the abstract. Could there be a little bit more variation?

Line 418: the authors state that “research has recently take a new direction focused on manipulating the splicing of these genes….”: in the review there is no mention of such studies. Please include.

Author Response

This article is written clearly, the contents are accurately and coherent, and the English level is very good. Several points could be addressed to make the text much more exciting and fluent to read:

The introductory part could be improved by giving more general and comprehensive information of VEGF-A, VEGFRs, neuropilins and the induced signaling path in the context of diverse tissue and organ settings. The first paragraph immediately goes too much into detail of listing signalling members, instead of attracting attention to the importance of this growth factor pathway. Evidence should be underscored by respective citations of additional key research papers / reviews. Maybe use text blocks included in subsequent sections of the manuscript.

Response: We have altered the introduction to give a more general overview of VEGF-A and the RTK signalling.

The second paragraph leads over to the splice variants of VEGFR-1 and VEGFR-2. The beginning of this section could be kept more introductory and general, maybe also by improving Figure 1

(e.g. include information about Ig-like domains).

Response: We have altered this paragraph to ensure it is more introductory and general.

Overall, the sub-headings and flow of the subsections could be improved by including more descriptive text in the headlines.

Response: We have added more descriptive text to the headings.

The very extensive “VEGR-1” section (line 160f) could be split into two parts “VEGFR-1 signaling” and “function of soluble sVEGR-1”.

Response: We have split this section as you suggested.

An overall summary and/or hints on the implications of the research for clinical application is not provided. Examples for therapeutic intervention with VEGF-A and VEGFR-splicing are missing, maybe drawing the attention to recent innovations, underlined by reviews.

Response: We have included a brief paragraph at the end of the manuscript with some examples of how research into the splicing of VEGF-A and its receptors could lead to potential therapeutic interventions.

Detailed comments:

Line 26f: please include respective citations throughout the whole first paragraph.

Line 29: be more explicative about the neuropilins “there are co-receptors in this axis…”

Line 35: maybe more extensive title such as “ VEGFR splice variants and functions”

Line 36f: refer to the protein structure shown in Figure 1.

Lines 41 and 92: maybe more extensive heading than just “VEGFR-1” and “VEGFR-2”?

Line 42: incomplete introductory sentence, which cell types / tissues? add according references.

Line 46: is the regulation by hypoxia ubiquitous in all tissues?

Line 51: unclear explanation, what is VEGFR-2 function? What does that mean for VEGFR-1 function?

Line 53: please give a brief summary of what is the hypothesized overall role of VEGR-1.

Line 55: conflict to line 36, where the authors mention seven Ig-like domains.

Line 58: unclear description of splicing process. What does mean “following intron 13 retention”?

Do you mean an exon 13+14 fusion?

Line 62: functioning of VEGFR-1 plus VEGFR-2 needs more explanation to understand effect on mouse rescue phenotype / reduction of VEGFR-2 phosphorylation.

Line 64: “there are currently five isoforms known

Line 69: maybe a brief hint, where these isoforms have been detected?

Line 71: “fully expressed transmembrane VEGFR-1”

Line 85f: legend to Figure 1: maybe better “alternative splice variants of VEGFR-1 and VEGFR-2”.

Line 89: B) instead of 2)

Line 102: include according reference, which proteolytic cleavage?

Line 120: what means “anti-permeability”? please use better description.

Line 122: please briefly explain what is meant with “translational read-through”.

Line 132: “upon phosphorylation of multiple se/arg repeats…”

Line 140: “…, thus they are suggested to regulate….”

Line 143: “…results in the longer isoforms…”

Line 144: “…lack exon 6 and 7, so they are unable to bind HS,…”

Line 148: “…regarding exon 8 of the VEGF-A gene, selection of either the proximal or the distal splice site has been reported to be dependent on the type of external stimulus.”

Line 152: please explain the full names of SRPK1 and Clk-1.

Lines 160: longer subtitle? Maybe “role of VEGFR-1 signaling and sVEGFR-1 isoforms”?

Line 161: “as mentioned above,…”

Line 228: longer subtitle? Maybe “mechanism of VEGFR-2 signaling”?

Line 291: longer subtitle? Maybe “role of sVEGFR-2”?

Line 300: “…isoforms act anti-angiogenic and anti-permeable…”

Line 354: “…when the VEGFR-2 abundance is higher”

Line 355: “there is evidence that suggests that..”

Line 362: better subtitle maybe “roles of neuropilins NRP1 and NRP2”

Line 363f: maybe better introductory sentence: “neuropilins can function as co-receptors of VEGFA, VEGFR-1 and VEGFR-2.”

Line 385: better “NRP1 and NRP2 splice variants”?

Line 388: “…in the NRP1 gene, resulting in proteins that….”

Line 392: what do you mean with “anti-tumor properties”?

Line 404: “…and govern different molecular pathways as a result.”

Line 407: “…with respect to VEGF-A binding and signaling”

Line 415f: this part of the conclusion is just repetition of the abstract. Could there be a little bit more variation?

Line 418: the authors state that “research has recently take a new direction focused on manipulating the splicing of these genes….”: in the review there is no mention of such studies. Please include.

Response: Thank you of your detailed comments. We have addressed each of these comments in the revised manuscript.

Reviewer 2 Report

This review article describes recent findings of the functions of VEGFRs and VEGF-A splice variants in the VEGF-A/VEGFR signaling axis. This review encompasses major previous works describing splicing variation and function of VEGF-A/VEGFR-1/VEGFR-2, and is very helpful for readers. So, the reviewer thinks this review is suitable for publication in this journal. However, the reviewer thinks that the authors' responses to the following comments would improve the paper. 

Minor points

1. Line 66: Isoform 2 of VEGFR-1 (sVEGFR-1) has a specific "31" amino acid residues in C-terminus. 

2. Line 79-80: "Several drugs" should be changed to "Several drugs and protein factors" because Jmjd6 and U2AF65 are proteins, and "sVEGFR-1/sVEGFR-1_i14 expression" should be changed to "sVEGFR-1/sVEGFR-1_i13 expression" because they detected sVEGFR-1_i13 using the specific primer for intron 13 in the analyses. 

Furthermore, when you mention the regulatory mechanism of the alternative splicing of VEGFR-1, you should cite the following paper. It showed another mechanism to regulate alternative polyadenylation of sVEGFR-1_i13.

Ikeda et al. Regulation of soluble Flt-1 (VEGFR-1) production by hnRNP D and protein arginine methylation. Mol. Cell. Biochem. 413, 155-164, 2016.

3. Line 144: VEGF-A111 is described in the manuscript although this variant is not included in Figure 2. You should include the VEGF-A111 variant in Figure 2.

4. Line 196-198: Reference [16] is not suitable for this context. If you want to mention the pre-eclampsia, you should cite the following paper. They showed that the sVEGFR-1_e15a isoform is a major contributor to the pathogenesis of pre-eclampsia.

Ashar-Patel et al. FLT1 and transcriptome-wide polyadenylation site (PAS) analysis in preeclampsia. Sci. Rep. 7, 1-14, 2017.

5. Line 203-207: If you want to mention that sVEGFR-1 is important for the function of glomerular epithelial cells (podocyte), you should cite the following paper.

Jin et al. Soluble FLT1 Binds Lipid Microdomains in Podocytes to Control Cell Morphology and Glomerular Barrier Function. Cell 151, 384-399, 2012.

6. Line 295: Reference [96] is not suitable because it does not report about sVEGFR-2.

Author Response

This review article describes recent findings of the functions of VEGFRs and VEGF-A splice variants in the VEGF-A/VEGFR signaling axis. This review encompasses major previous works describing splicing variation and function of VEGF-A/VEGFR-1/VEGFR-2, and is very helpful for readers. So, the reviewer thinks this review is suitable for publication in this journal. However, the reviewer thinks that the authors' responses to the following comments would improve the paper.

Minor points

1. Line 66: Isoform 2 of VEGFR-1 (sVEGFR-1) has a specific "31" amino acid residues in C-terminus.

Response: Thanks. We have corrected this.

2. Line 79-80: "Several drugs" should be changed to "Several drugs and protein factors" because Jmjd6 and U2AF65 are proteins, and "sVEGFR-1/sVEGFR-1_i14 expression" should be changed to "sVEGFR-1/sVEGFR-1_i13 expression" because they detected sVEGFR-1_i13 using the specific primer for intron 13 in the analyses.

Response: We have corrected this. By sVEGFR-2, we mean the il3 isoform.

Furthermore, when you mention the regulatory mechanism of the alternative splicing of VEGFR-1, you should cite the following paper. It showed another mechanism to regulate alternative polyadenylation of sVEGFR-1_i13.

Ikeda et al. Regulation of soluble Flt-1 (VEGFR-1) production by hnRNP D and protein arginine methylation. Mol. Cell. Biochem. 413, 155-164, 2016.

Response: We have added this paper.

3. Line 144: VEGF-A111 is described in the manuscript although this variant is not included in Figure 2. You should include the VEGF-A111 variant in Figure 2.

Response: We have added it to Figure 2.

4. Line 196-198: Reference [16] is not suitable for this context. If you want to mention the pre-eclampsia, you should cite the following paper. They showed that the sVEGFR-1_e15a isoform is a major contributor to the pathogenesis of pre-eclampsia.

Ashar-Patel et al. FLT1 and transcriptome-wide polyadenylation site (PAS) analysis in preeclampsia. Sci. Rep. 7, 1-14, 2017.

Response: Apologies for our mistake. We have corrected this.

5. Line 203-207: If you want to mention that sVEGFR-1 is important for the function of glomerular epithelial cells (podocyte), you should cite the following paper.

Jin et al. Soluble FLT1 Binds Lipid Microdomains in Podocytes to Control Cell Morphology and Glomerular Barrier Function. Cell 151, 384-399, 2012.

Response: Thank you. We have added it to the manuscript.

6. Line 295: Reference [96] is not suitable because it does not report about sVEGFR-2.

Response: Apologies for our mistake. We have corrected this.